# Early Biofilm Formation on the Drain Tip after Total Knee Arthroplasty Is Not Associated with Prosthetic Joint Infection: A Pilot Prospective Case Series Study of a Single Center

**DOI:** 10.3390/healthcare12030366

**Published:** 2024-01-31

**Authors:** Marco Grassi, Marco Senarighi, Luca Farinelli, Annamaria Masucci, Monica Mattioli-Belmonte, Caterina Licini, Antonio Gigante

**Affiliations:** 1Clinical Orthopedics, Department of Clinical and Molecular Science, School of Medicine, Università Politecnica Delle Marche, 60121 Ancona, Italysenarighi.m@gmail.com (M.S.); a.p.gigante@staff.univpm.it (A.G.); 2Laboratory of Clinical Pathology and Microbiology, General Service Department, Azienda Ospedaliera Universitaria “Ospedali Riuniti”, 60121 Ancona, Italy; annamaria.masucci@ospedaliriuniti.marche.it; 3Department of Clinical and Molecular Sciences, Università Politecnica Delle Marche, 60121 Ancona, Italy; m.mattioli@staff.univpm.it (M.M.-B.); c.licini@staff.univpm.it (C.L.)

**Keywords:** periprosthetic joint infection, total knee arthroplasty, infection disease, biofilm, surgical drainage

## Abstract

Background: Periprosthetic joint infection (PJI) is a devastating complication of arthroplasties that could occur during the surgery. The purpose of this study was to analyze the biofilm formation through microbiological culture tests and scanning electron microscopy (SEM) on the tip of surgical drainage removed 24 h after arthroplasty surgery. Methods: A total of 50 consecutive patients were included in the present prospective observational study. Drains were removed under total aseptic conditions twenty-four hours after surgery. The drain tip was cut in three equal parts of approximately 2–3 cm in length and sent for culture, culture after sonication, and SEM analysis. The degree of biofilm formation was determined using a SEM semi-quantitative scale. Results: From the microbiological analysis, the cultures of four samples were positive. The semi-quantitative SEM analysis showed that no patient had grade 4 of biofilm formation. A total of 8 patients (16%) had grade 3, and 14 patients (28%) had grade 2. Grade 1, scattered cocci with immature biofilm, was contemplated in 16 patients (32%). Finally, 12 patients (24%) had grade 0 with a total absence of bacteria. During the follow-up (up to 36 months), no patient showed short- or long-term infectious complications. Conclusions: Most of the patients who underwent primary total knee arthroplasty (TKA) showed biofilm formation on the tip of surgical drain 24 h after surgery even though none showed a mature biofilm formation (grade 4). Furthermore, 8% of patients were characterized by a positivity of culture analysis. However, none of the patients included in the study showed signs of PJI up to 3 years of follow-up.

## 1. Introduction

Closed-suction drainages are widely used after elective or emergency orthopedic surgery to prevent hematoma formation and local fluid accumulation, reducing limb swelling and promoting wound healing [1,2]. Recently, the systematic use of drains after orthopedic procedures has been questioned [3,4]. Hence, a recent systematic review reported that the routine use of drains does not add any benefits after total knee arthroplasty (TKA) [5]. Moreover, a recent randomized controlled trial reported that the application of intraarticular suction drainage following TKA did not show any effects on postoperative hemoglobin levels, blood loss, prosthetic knee range of motion, Knee Injury and Osteoarthritis Outcome Score (KOOS), or the length of hospital stay. Conversely, the authors observed that drains were associated with a greater complication rate: infection and knee stiffness. Therefore, they concluded that the usage of a suction drain may not provide any benefit and may have adverse effects in the TKA procedure [6].

Indeed, there is increasing evidence that suction drains may increase the incidence of local wound complications and infections, especially if they remain in situ for more than twenty-four hours [7,8]. Moreover, microbial agents are known to form biofilm on the surface of several devices (e.g., drains or prostheses), improving resistance to antimicrobial agents. Furthermore, materials such as hydroxyapatite and other bioactive substances may be more prone to bacterial adhesion than bioinert metals such as titanium or stainless steel [9].

Periprosthetic joint infection (PJI) is a devastating complication of arthroplasties. The diagnosis of PJI still remains difficult, particularly in its acute postoperative stage due to postoperative inflammation [10]. Early infections are mostly due to the contamination of the surgical site during surgery or in the first postoperative days [11,12]. The International Consensus Meeting on Musculoskeletal Infection (2018) stated that bacterial biofilm formation is a continuum and could be formed after just one day. The use of next generation sequencing analytical technology demonstrates a joint microbiome both in native joints (the microbiome of osteoarthritis) and prosthetic joint environments where, in a negative clinical setting, there is no PJI [13]. The systematic culture of the drain tip has been addressed in many studies, but the prognostic factor of the culture result after an elective orthopedic surgery is still unclear [14,15]. Dower et al. found that biofilm in closed-suction drains is already formed in the second postoperative hour in patients undergoing breast augmentation surgery [16]. Akiyama et al. inoculated *Staphylococcus aureus* onto animal wounds. They found the development of a cluster of cells (characteristic of a biofilm) after 6–24 h post inoculation [17]. To the best of our knowledge, there are no studies evaluating biofilm formation after total knee replacement. The primary aim of the present prospective observational study was to determine the biofilm formation on the tip of the drain of TKA removed 24 h after surgery. The secondary aim was to analyze whether any biofilm formation on the drain tip or its possible positivity on cultural examinations was associated with the development of PJI at a minimum follow-up of 3 years.

## 2. Materials and Methods

The study was designed as a prospective study of a single center. Fifty consecutive patients that underwent primary TKA at Clinical Orthopedics, Department of Clinical and Molecular Sciences, Università Politecnica delle Marche, from November 2019 to June 2020 were prospectively included in the present study if they fulfilled the following inclusion criteria: diagnosis of primary symptomatic knee osteoarthritis, age 60–80 years, and Kellgren–Lawrence (KL) grade IV. Patients with a previous history of knee infection, rheumatic disease, and/or systematic or local infection (i.e., cystitis or pneumonia in the previous 6 months) were excluded. Patients with surgical time greater than 90 min (from incision to closure) were excluded. All patients received the study protocol and provided their consent. The study was approved by the Internal Review Board (2022-377). All procedures followed the ethical standards of the responsible committee on human experimentation (institutional and national), and were in accordance with the Helsinki Declaration of 1975, as revised in 2008. Ethical compliance was obtained according to the Italian legislative decree of 14 May 2019, n. 52, containing amendments to the legislative decree of 6 November 2007, n. 200, implementing Directive 2005/28/EC. Patients were evaluated pre- and postoperatively (3 years of follow-up). Oxford knee score (OKS), pain (VAS), and any sign of PJI were noted.

### 2.1. Surgical Technique

The senior surgeon performed all surgeries. TKAs were performed using the medial parapatellar approach, and a cemented cruciate-retaining or posterior-stabilized total knee prosthesis (Persona knee prothesis, Zimmer Inc., Warsaw, IN, USA) was used. The tourniquet was routinely used from the beginning of the surgery until the end of cementation. Prophylactic antibiotics (i.e., 2 g of cefazoline) were routinely administered 30 min before incision. A 16-inch suction drain was routinely placed intra-articularly. All patients were given a standardized postoperative rehabilitation program that consisted of a four-point gait pattern within the first 2 weeks after surgery. Crutches were used for the initial 4 weeks. For the subsequent 8 weeks, low-impact physical activities such as walking, swimming, and static cycling were recommended.

### 2.2. Drain Removal, Biofilm Assessment, and Culture Analysis

The polyurethane drains were removed twenty-four hours after surgery by the same team (MG, MS, and LF). Before removal, a meticulous asepsis of the skin around the drain was performed with a 10% aqueous povidone-iodine solution. If the tip of the drain inadvertently came into contact with the skin, the patient was excluded from the study. The drain tip was cut into three equal parts of approximately 2–3 cm in length, maintaining the aseptic conditions. Three different specimens of the same drain were obtained (Figure 1). One specimen was sectioned longitudinally into two halves, appropriately fixed, and then prepared for SEM analysis. The other two specimens were collected in two different sterile containers and analyzed, respectively, using routine microbiological drain tip culture and sensitive broth culture after drainage sonication.

### 2.3. Microbiological Culture Analysis

The specimens intended for microbiological analysis were promptly stored in sterile containers and delivered to the microbiology laboratory of “Azienda Ospedaliera-Universitaria delle Marche” within 30 min. A specimen for each drain underwent a sonication procedure in order to improve the sensitivity of microbiological analysis due to the attachment of biofilm on the drain’s surface.

Sonication procedure: the contents of the drainage tip were opened under a laminar flow hood, the components were covered for at least 90% of their volume with sterile physiological solution or Ringer’s solution, the container was closed, and the sonication bath (BactoSonic-Bandelin, BANDELIN, Berlin, Germany) was prepared by filling the tub with sterile water. The container was vortexed for 30 s, and sonicated at 30–40 kHz 0.22 ± 0.04 W/cm^2^ for 5 min. A total of 0.1 mL of sonicate was inoculated into plates of Columbia Blood Agar, Chocolate Agar, and Mac Conkey Agar. The plates were incubated in a 5% CO_2_ atmosphere at 37 °C for 4 days and Mac Conkey Agar at 37 °C for 48 h. Then, 0.1 mL was inoculated in Blood Agar and the incubation time was extended until 14 days at 37 °C in anaerobiosis. To identify bacteria and fungi, a protein profile obtained using matrix-assisted laser desorption ionization-time of flight (MALDI-TOF MS, Biomerieux Italia, Grassina, Italy) mass spectrometry was used. The mass peaks achieved by the test strains were compared to those of known reference strains. Currently, it is possible to distinguish an organism from an isolate within a short time frame. For antimicrobial susceptibility testing, the automatic instrumentation VITEK-2 Biomerieux Italia or the broth microdilution test SENSITITRETM Thermofisher (Thermo Fisher Scientific, Waltham, MA, USA) was used. A second specimen of drain underwent culture analysis. Hence, the tip of the drain was unrolled on a plate, with the aid of sterile tweezers. The same culture media were used, with the same incubation methods as described above.

### 2.4. Scanning Electron Microscopy Analysis

The samples intended for SEM analysis were fixed in 2.5% glutaraldehyde (MERCK, Rahway, NJ, USA) in 0.1M sodium cacodylate buffer (Sigma, St. Louis, MO, USA, C-0250) for 1 h and then rinsed with 0.1 M sodium cacodylate buffer solution. After osmium tetroxide (Electron Microscopy Sciences, Hatfield, PA USA) fixation, another cacodylate buffer washing was performed and complete dehydration was achieved in graded alcohol series and hexamethyldisilane. The samples were then mounted on aluminum stubs, gold-sputtered, and observed with an SEM Philips XL 20 (FEI, Milan, Italy) at an accelerating voltage of 20 kV at 1500, 2000, and 4000 magnification (Figure 2). The degree of biofilm formation was determined using a semi-quantitative grading scale (Table 1 and Figure 3, Figure 4, Figure 5 and Figure 6). For the SEM analysis, the samples were longitudinally cut in half and both surfaces were observed. From each patient, five different images were taken at 500× magnification. Two reviewers (MMB and CL) independently reviewed all scans, and differences were resolved by a third reviewer (AG), who independently reviewed all cases of disagreement.

### 2.5. Statistical Analysis

Data were collected and organized using Excel (v.16.77.1) (Microsoft, Redmond, WA, USA). Categorical variabilities were expressed in numbers and percentages. Continuous variabilities were expressed as averages and standard deviation (DS) or median and interquartile range (IQR) for normally and non-normal distributed data, respectively. The Shapiro–Wilk test was used to assess the normality of continuous data. All continuous variables were found to be non-normal, indicating an appropriate nonparametric approach. The non-parametric Friedman tests were used for comparing pre- and postoperative data. All statistical analyses were performed using SPSS (Version 28.0.1, IBM Corp., Armonk, NY, USA) A *p* < 0.05 was considered as significant. A prior power analysis was performed with G-Power test. Considering the F test, one tail, an effect size of 0.3, an alfa error of 0.5, and a power of study of 95%, 28 patients were required for the study.

## 3. Results

### 3.1. Microbiological Results

From the microbiological analysis, four (8%) patients were characterized by culture positivity; specifically, two (4%) patients resulted positive for *Staphylococcus epidermidis* from the cultural analysis of drain’s specimen without sonication, whereas two (4%) patients were positive for *Enterococcus faecium* from the cultural test after the sonication procedure. None of these patients were positive for both analyses. No polymicrobial cultures were detected. From the SEM analysis, it was observed that the same samples were characterized by biofilm formation. Specifically, the two (4%) patients with *Enterococcus faecium* positivity had a biofilm score of grade 1, whereas those with *Staphylococcus epidermidis* positivity had a score grade of 3. The semi-quantitative analysis using SEM showed that no samples were characterized by grade 4 of biofilm formation. A total of 8 (16%) patients had grade 3 and 14 (28%) patients had grade 2. Grade 1, scattered cocci with immature biofilm, was contemplated in 16 (32%) patients. Finally, in 12 (24%) patients, a total absence of bacteria was reported (grade 0).

### 3.2. Clinical Results

Patients’ characteristics are reported in Table 2. The mean follow up was 40 months, ranging from 38 to 45 months. All patients reported a significant clinical improvement in terms of OKS and VAS after surgery at a minimum of 3 years of follow-up (*p* < 0.05) (Table 3). No patients characterized by culture positivity and/or biofilm formation reported superficial wound infection, fistula, or any other sign of PJI during the 3-year follow-up period.

## 4. Discussion

The most important finding of the present study was that 76% of patients (38 of 50) who underwent primary TKA showed biofilm formation on the tip of the surgical drain 24 h after surgery, even though none showed a mature biofilm formation (grade 4). Furthermore, 8% of patients were characterized by culture positivity. However, none of the patients included in the study showed signs of PJI up to 3 years of follow-up. From our results, it can be noted that patients with a positive culture showed no concordance between the conventional analysis and culture performed after sonication. Hence, it should be expected that conventional culture positivity would also lead to culture positivity after sonication. A possible explanation was that a contamination could lead to culture positivity with conventional analysis, but not after sonication due to the non-formation of biofilm. On the other hand, the positivity after sonication could not correspond to positivity with conventional analysis due to the low sensitivity of the analysis or the senescence of the biofilm [18].

Our results were in line with previous research. Using next generation sequencing technology, Tarabichi M. et al. reported the presence of microbes in 35.3% of primary arthroplasties (hip and knee) without any sign of PJI, demonstrating the presence of microbiome in prosthetic joint environments [13]. Furthermore, Dower et al. found that biofilm in closed-suction drains is already formed in the second postoperative hour in patients undergoing breast augmentation surgery [16]. Sambanthamoorthy et al. found a mature *Staphylococcus aureus* biofilm formation at 12 h in an in vitro flow-cell study [19]. It is imperative to consider that bacterial colonization and the formation of biofilm represent a common complication on invasive devices [20,21]. Indeed, drawing a comparison with urologic surgery, the urinary catheters used to empty the bladder and collect urine in a drainage bag could be subject to bacterial colonization. Several studies reported that biofilms could readily develop on the inner or outer surfaces of urinary catheters upon insertion [22]. Analogously to our results, the dominant organisms isolated from urinary catheters were *S. epidermidis*, *S. aureus*, *Enterococcus faecalis*, and *Escherichia coli* [23]. However, as reported in previous studies, it should be considered that colonization and infection remain two different processes; therefore, the isolation of microorganisms from urinary catheters does not mean infection [24]. Indeed, in orthopedic surgery, several studies showed that the use of drainage did not significantly increase the risk of wound or deep infection after TKA [25,26]. In light of our results, no patients showed symptoms and signs of PJI at 3 years of follow-up. On the other hand, a significant clinical improvement in terms of OKS and VAS was observed. However, it remains to be elucidated if the presence of biofilm might be predictive of a higher risk of PJI.

### Limitations

The present study presents limitations that warrant disclosures. Firstly, a semi-quantitative grading scale was used for the SEM evaluation of biofilm formation. However, no other methods have been found in the literature and the focus of the present study was to assess the presence or not of biofilm on the drain tip. The semi-quantitative scale was only the method available to report the results. A quantitative molecular approach such as real-time PCR would have been the best method for the analysis of the presence of bacteria, but the substantial costs did not allow us to conduct it. The sample size was limited, but this reflects the pilot nature of the study, considering also the high costs of SEM analyses. The follow-up was limited.

## 5. Conclusions

In our population of 50 patients who underwent primary TKA, 76% showed immature or partially mature biofilm formation on the tip of the surgical drain at 24 h from surgery. Instead, 8% of patients were characterized by a culture positivity of the surgical drain. No patients showed symptoms or signs of PJI at 3 years of follow-up. Further studies are needed to establish whether the formation of biofilm and/or the culture positivity determine a higher risk of future PJI, or they only represent a contamination.

## Figures and Tables

**Figure 1 healthcare-12-00366-f001:**
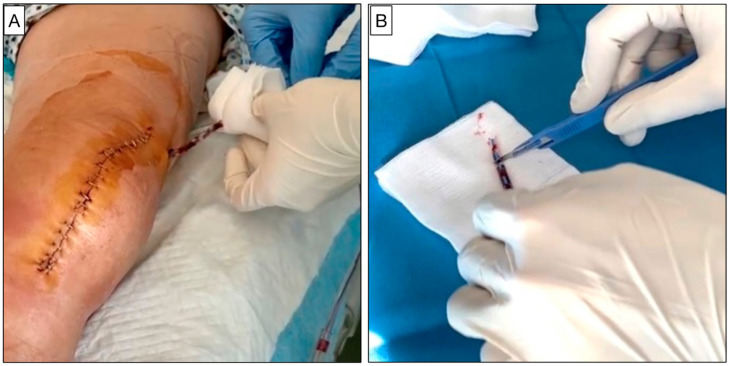
(**A**) The removal of the surgical drain on the first day after TKA surgery performed in a sterile and standardized manner. (**B**) Surgical drain tip section on a sterile surface.

**Figure 2 healthcare-12-00366-f002:**
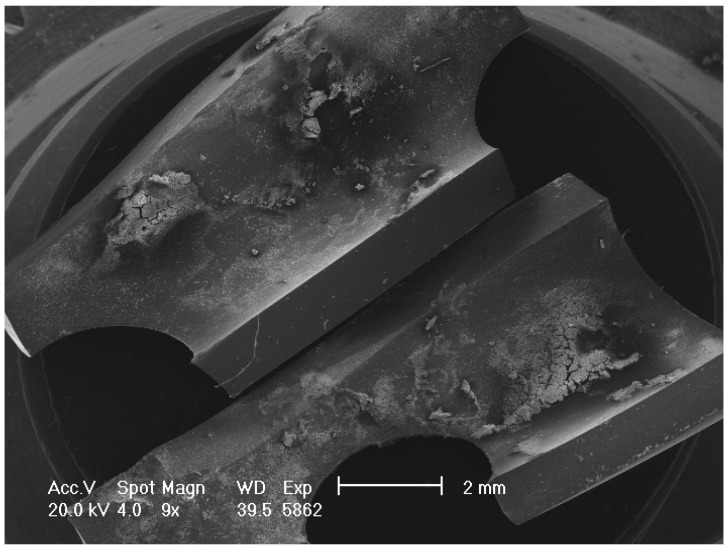
Low magnification of SEM image showing the tip of the drain sectioned longitudinally.

**Figure 3 healthcare-12-00366-f003:**
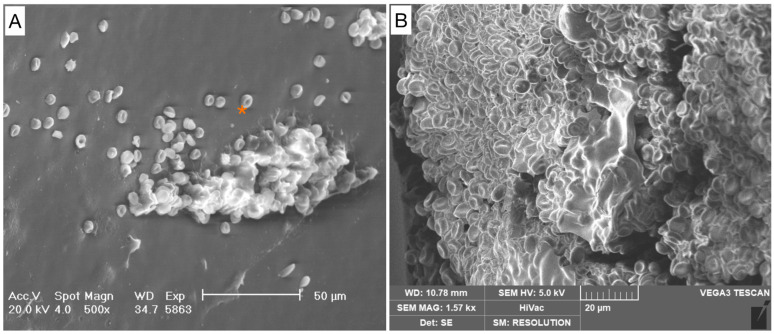
Grade 0. Absence of pathogenic cells. (**A**) Scattered red blood cells (*) are visible in the background. No pathological cells are detectable. (**B**) Blood clot.

**Figure 4 healthcare-12-00366-f004:**
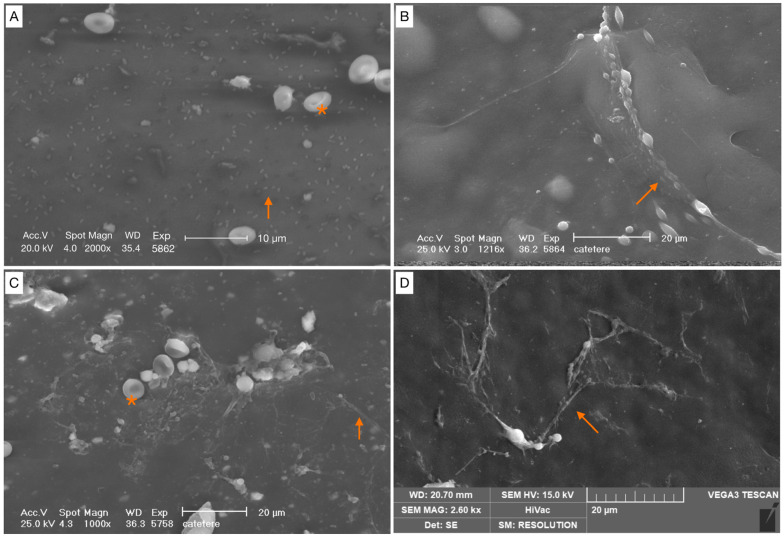
Scattered cocci, immature biofilm (covering < 25% of the specimen surface). (**A**) Bacterial cells appear irregular, without a matrix covering (orange arrow). (**B**–**D**) Cells were arranged either as individual cells, or as short chains with a filamentous projection of extracellular matrix (orange arrows). Red blood cells (*).

**Figure 5 healthcare-12-00366-f005:**
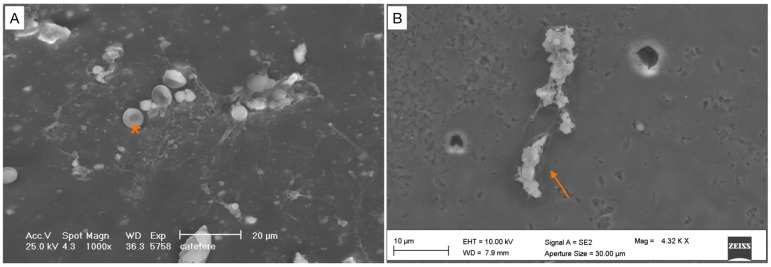
Partially mature biofilm (covering < 50% of the specimen surface). (**A**) Bacterial cells are arranged in a cluster, with extracellular matrix in the background. (**B**) High magnification. Bacterial cells are arranged in a cluster (orange arrow). Red blood cells (*).

**Figure 6 healthcare-12-00366-f006:**
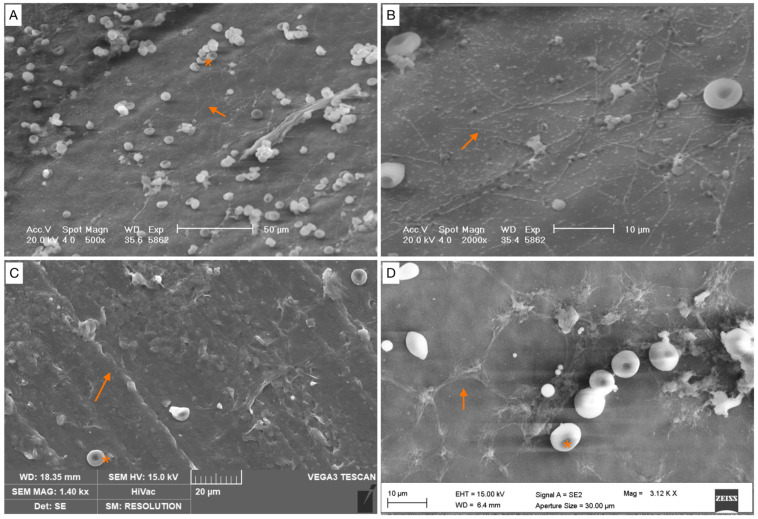
Mature biofilm (covering between 50 and 75% of the specimen surface). (**A**–**D**) Images of biofilm structure. Biofilms show a cobwebbed appearance, with an amorphous polymeric extracellular matrix surrounding (orange arrow) and interconnecting bacteria. Red blood cells (*).

**Table 1 healthcare-12-00366-t001:** Semi-quantitative scale of biofilm formation (Appendix A).

Grade	Biofilm Formation
0	Absence of pathogenic cells
1	Scattered cocci, immature biofilm (covering < 25% of the specimen surface)
2	Partially mature biofilm (covering < 50% of the specimen surface)
3	Mature biofilm (covering between 50 and 75% of the specimen surface)
4	Mature biofilm (covering the entire specimen surface)

**Table 2 healthcare-12-00366-t002:** Patients’ characteristics.

Variables		Cohort (*n* = 50)
Gender		
	Female, *n* (%)	16 (32%)
	Male, *n* (%)	34 (68%)
Age (years)		
	Mean ± SD	72.1 ± 6.3
	Range	55–87
Body mass index (BMI)		
	Mean ± SD	25.8 ± 4.2
	Range	20.4–35.1
Smoking		
	Yes/no	6/44
Diabetes		
	Yes/no	3/47

**Table 3 healthcare-12-00366-t003:** Pre- and post-operative clinical results. OKS: Oxford knee score. VAS: visual analogue scale.

Variables	Pre-Operatively	Three Years of Follow-Up	*p*-Value
OKS			
Median (IQR, 25th–75th percentile)	25 (9, 22–31)	43 (5, 40–45)	<0.05
VAS			
Median (IQR, 25th–75th percentile)	6 (4, 4–8)	0 (1, 0–1)	<0.05

## Data Availability

Data are unavailable due to privacy and ethical restrictions.

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
