# Peer review of "Early Biofilm Formation on the Drain Tip after Total Knee Arthroplasty Is Not Associated with Prosthetic Joint Infection: A Pilot Prospective Case Series Study of a Single Center"

_healthcare, 2024, doi:10.3390/healthcare12030366_

Round 1

Reviewer 1 Report (Previous Reviewer 2)

Comments and Suggestions for Authors

I have carefully examined each of the images provided by the authors. Unfortunately, the anatomical definition of biofilm is an exopolymeric matrix in which bacteria that produced it are embedded. This definition cannot be extracted from the images provided by the authors. What the authors describe as a matrix could very well be the layer of plasma proteins that forms within seconds of any foreign body coming into contact with the blood. With the fixation methods described in the article, the image of a mature biofilm would be bacterial cells between 0.3 to 0.5 um for streptococci or 0.5-0.8 um for staphylococci forming at least two or three layers and held together by strands of exopolysaccharides where there may or may not be plasma (cellular or acellular) components. Figures 5 and 6 do not represent a mature biofilm. The most worrying thing of all is that the only clear example where the clear presence of bacteria is detected presents a bacilliform bacterium (Figure 4A), presumably Enterobacteriaceae, a microorganism that was not detected in the patients' cultures according to the results described in the study itself.

For me, the results are inconclusive and I am unable to recommend this article for publication. 

Comments on the Quality of English Language

Minor editing of English language required

Author Response

please see attached

Reviewer 2 Report (New Reviewer)

Comments and Suggestions for Authors

In this paper, the authors conducted a microbiological analysis of patients undergoing a knee replacement surgery. They found that out of the 50 patients examined, 4 (8%) were culture-positive either with or without sonication but none of these patients tested positive in both analyses, and no polymicrobial cultures were detected. The authors also conducted scanning electron microscopy (SEM) analysis, which revealed various degrees of biofilm formation. Specifically, 8 patients (16%) had grade 3, 14 (28%) patients had grade 2, 16 (32%) patients had grade 1, 12 (24%) patients had grade 0 (a total absence of bacteria), and none of the samples had grade 4 of biofilm formation. Furthermore, none of the patients experienced superficial wound infections, fistulas, or any other signs of PJI during the 3-year follow-up period. Overall, this paper is well written and the presentation of results is great. I only have the following minor comments for the authors to address:

1. End of line 26 ("...none showed a mature biofilm formation (grade IV)."): To stay consistent with the rest of notations, I'd suggest using "grade 4".

2. Line 73 ("Fifty consecutive patients that underwent to primary TKA...") and other places: Please remove the word "to" after "underwent".

3. Line 177 ("range (38-45)"): I would suggest rephrasing this in words, i.e., "ranging from 38 to 45 months".

4. Lines 180-181 ("at 3-year of follow-up"): This phrase doesn't seem right. I'd suggest revising it to "during the 3-year follow-up period" or whatever makes sense.

Comments on the Quality of English Language

English is generally fine.

Author Response

please see attacched

Reviewer 3 Report (New Reviewer)

Comments and Suggestions for Authors

Author Response

please see attacched

This manuscript is a resubmission of an earlier submission. The following is a list of the peer review reports and author responses from that submission.

Round 1

Reviewer 1 Report

Comments and Suggestions for Authors

Very interesting study examining potential relationship between early biofilm formation on the drain tip after TKA and PJI. Outcomes based on 50 patients demonstrated that although the percentage of the biofilm formation from the drain tip among patients who had received TKA was high, no patient had developed PJI. This study, falls within the scope of the journal, is presented well.

I have only one comment regarding the removal of the drains under total aseptic conditions. How could the authors confirm the conditions around the drain is totally aseptic even if they had used 10% aqueous povidone-iodine solution? If possible, I would suggest swabbing the skin, especially for the skin surrounding the drain as a control.

Please provide the full names for any abbreviations for the first appearance.

Comments on the Quality of English Language

Minor revision.

Reviewer 2 Report

Comments and Suggestions for Authors

The study is a prospective observational study with the aim of determining the biofilm formation in the tip of the drain of total knee arthroplasty (TKA) removed 24 hours after surgery and their relationship with the potential development of prosthetic joint infection (PJI) inside a minimum follow-up of 3 years. The study presents serious flaws that make it difficult to support the conclusion reached in it.

Major comment

Lines 115-131: In how much volume were the drain tips sonicated? Was there no centrifugation of the sonication solution after the sonication?

Table 1.: Could the author justify based on which criteria they decided the grades reflected in Table 1? What does “specimen surface” mean? How many pictures were taken from each drain tip and at which magnification?

Could the author show all the pictures from each patient’s sample as Supplemental results? What about the pictures from the positive samples?

Considering the pictures contained in Figure 3, it is extremely hard to believe in the sensitivity of the SEM study. Why is Figure 3b showing bacilli onto the drain tip if no Gram-negative bacilli PJI was informed in the study? If we observe the 20-um scale bar from Figure 3c, the “cocci” observed must have a minimum diameter of 5 um. There are no Gram-positive cocci with that size. The normal diameter for a staphylococcus ranges between 0.5 and 0.8 um and for a streptococcus ranges between 0.3 and 0.5 um.

The study is based on the fact that there is a microbiota in the prosthetic joint environment, but this link cannot be made because both methodological approaches are categorically opposed. The SEM study lacks sensitivity, this kind of study is often made with the intention of illustrating punctually the biofilm growth/development or its distribution onto a specific surface, not for quantifying the covered surface. On the contrary, the microbiome-related approach is extremely sensitive in detecting any microorganism whatever its metabolic state. The only way of correctly supporting what the authors have intended is having determine the presence or absence of bacteria in the drain tip using a quantitative molecular approach. There is no plausible and parsimonious explanation for why the bacteria coming from a “young” bacterial biofilm cannot grow and be detected through sonication.

Minor comments

Line 53: The “acute infection” definition is wrong. A haematological PJI is also acute by definition.

Throughout the manuscript, there are some words and expressions that must be in italics, e.g., et al. and the bacterial scientific names (line 63).

Table 1 is lacking in a table caption.

Comments on the Quality of English Language

Moderate editing of the English language would be required.

Reviewer 3 Report

Comments and Suggestions for Authors

Dear authors,

In this paper you sought to determine the biofilm formation in the tip of the drain of knee arthroplasty. Please find specific comments below.

General comments

The paper reads well but there are shortcomings that need to be addressed by the authors. First of all, I believe the discussion section need to be improved and extended.

Abstract: Please note you should not be reporting numbers in the conclusion section of the abstract (line 24). Instead, a general conclusion will suffice.

Provide the IRB number please in the methods section.

Did you account for the cruciate retaining and PS design as in conducting a subgroup analysis?

Also, how was sample size calculated? Are 50 patients enough to test your hypothesis? You have mentioned this limitation in the corresponding discussion section, but I am afraid you have not followed any pilot study guidelines to conduct this paper. For example, there are papers describing how to appropriately report a pilot study such as the one shown below.

Reference

Teresi JA, Yu X, Stewart AL, Hays RD. Guidelines for Designing and Evaluating Feasibility Pilot Studies. Med Care. 2022 Jan 1;60(1):95-103. doi: 10.1097/MLR.0000000000001664. PMID: 34812790; PMCID: PMC8849521.

Comments on the Quality of English Language

None

Round 2

Reviewer 2 Report

Comments and Suggestions for Authors

The study is a prospective observational study with the aim of determining the biofilm formation in the tip of the drain of total knee arthroplasty (TKA) removed 24 hours after surgery and their relationship with the potential development of prosthetic joint infection (PJI) inside a minimum follow-up of 3 years. The study still raises some doubts that make it difficult to support the conclusion reached in it.

The criteria on which they decided the grades reflected in Table 1 is not clear. At those magnifications, you cannot distinguish bacterial biofilm. Furthermore, the representative images are not clear at all and the author denied choosing better ones or sharing them as Supplemental data.

Altogether, I am obliged to reject the manuscript as it is for publication.

Comments on the Quality of English Language

Minor editing of English language required.

Reviewer 3 Report

Comments and Suggestions for Authors

Dear authors,

You have now addressed my original comments and I believe the paper is publishable.

Comments on the Quality of English Language

-